# A Randomised-Controlled Clinical Study Examining the Effect of High-Intensity Laser Therapy (HILT) on the Management of Painful Calcaneal Spur with Plantar Fasciitis

**DOI:** 10.3390/jcm10214891

**Published:** 2021-10-23

**Authors:** Piotr Tkocz, Tomasz Matusz, Łukasz Kosowski, Karolina Walewicz, Łukasz Argier, Michał Kuszewski, Magdalena Hagner-Derengowska, Kuba Ptaszkowski, Robert Dymarek, Jakub Taradaj

**Affiliations:** 1Institute of Health Sciences, University of Opole, 45-060 Opole, Poland; piotrtkocz@op.pl (P.T.); tomaszmatusz@icloud.com (T.M.); lukaszkosowski88@gmail.com (Ł.K.); karolina.w101@wp.pl (K.W.); argier@o2.pl (Ł.A.); 2Institute of Physiotherapy and Health Sciences, Academy of Physical Education, 40-065 Katowice, Poland; m.kuszewski@awf.katowice.pl (M.K.); j.taradaj@awf.katowice.pl (J.T.); 3Faculty of Earth Sciences and Spatial Management, Nicolaus Copernicus University, 87-100 Torun, Poland; m.hagner-derengowska@umk.pl; 4Department of Physiotherapy, Wroclaw Medical University, 50-355 Wroclaw, Poland; kptaszkowski@gmail.com; 5Department of Rehabilitation, TOMMED Medical Center, 40-662 Katowice, Poland

**Keywords:** high-intensity laser therapy, calcaneal spur, plantar fasciitis, pain management, Visual Analogue Scale, Laitinen Pain Scale

## Abstract

Calcaneal spur and plantar fasciitis are the most common causes of plantar heel pain. There are many effective physical modalities for treating this musculoskeletal disorder. So far, the are no clear recommendations confirming the clinical utility of high-intensity laser therapy (HILT) in the management of painful calcaneal spur with plantar fasciitis. This study aimed to evaluate the effectiveness of HILT in pain management in patients with calcaneal spur and plantar fasciitis. A group of 65 patients was assessed for eligibility based on the CONSORT guidelines. This study was prospectively registered in the Australian New Zealand Clinical Trial Registry platform (registration number ACTRN12618000744257, 3 May 2018). The main eligibility criteria were: cancer, pregnancy, electronic and metal implants, acute infections, impaired blood coagulation, cardiac arrhythmias, taking analgesic or anti-inflammatory medications, non-experience of heel pain, or presence of other painful foot conditions. Finally, 60 patients were randomly assigned into two groups: study group (*n* = 30, mean age 59.9 ± 10.1), treated with HILT (7 W, 149.9 J/cm^2^, 1064 nm, 4496 J, 12 min), and placebo-controlled group (*n* = 30, mean age 60.4 ± 11.9), treated with sham HILT therapy. Both groups received ultrasound treatments (0.8 W/cm^2^, 1 MHz frequency, 100% load factor, 5 min). Treatment procedures were performed once a day, five times per week for three weeks (total of 15 treatment sessions). Study outcomes focused on pain intensity and were assessed before (M1) and after (M2) the treatment as well as after 4 (M3) and 12 (M4) weeks using the Visual Analogue Scale (VAS) and the Laitinen Pain Scale (LPS). According to VAS, a statistically significant decrease in the study group was observed between M1 and M2 by 3.5 pts, M1 and M3 by 3.7 pts, and M1 and M4 by 3.2 pts (*p* < 0.001). On the other hand, the control group showed a statistically significant decrease (*p* < 0.001) between M1 and M2 by 3.0 pts, M1 and M3 by 3.4 pts, and M1 and M4 by 3.2 pts. According to LPS, a statistically significant decrease in the study group was observed between M1 and M2 by 3.9 pts, M1 and M3 by 4.2 pts, and M1 and M4 by 4.0 pts (*p* < 0.001). On the other hand, the control group showed a statistically significant decrease between M1 and M2 by 3.2 pts (*p* = 0.002), M1 and M3 by 4.0 pts (*p* < 0.001), and M1 and M4 by 3.9 pts (*p* < 0.001). However, there were no statistically significant differences between the groups in VAS and LPS (*p* > 0.05). In conclusion, the HILT does not appear to be more effective in pain management of patients with calcaneal spurs and plantar fasciitis than the conservative standard physiotherapeutic procedures.

## 1. Introduction

A thorough analysis of the abundant literature addressing high-intensity laser therapy (HILT) at the cellular and tissue level (experimental work, in vitro, and animal experiments) reveals several interesting and documented developments that may lay a plausible foundation for therapeutic mechanisms in many disease entities [1,2,3,4,5,6]. 

Laser irradiation offers a specific dose of energy (photons) to the areas of the tissue to be treated. Researchers demonstrate the effect of laser beam at the cellular level is manifested by increased production of ATP, increased activity of membrane enzymes, increased synthesis of DNA and RNA, and acceleration of electrolyte exchange between the cell and the surrounding areas. At the tissue level, acceleration of blood and lymph circulation, reduced intracapillary pressure, increased excitability threshold of nerve endings, and stimulation of immune response are observed. The phenomena described above constitute the basis for the described analgesic and anti-inflammatory mechanisms [7,8].

It was also confirmed by a large number of clinical papers that demonstrate the usefulness of HILT in terms of musculoskeletal disorders [9,10,11,12]. However, it should be noted that there are still some reports that are critical of laser therapy [13,14,15]. 

Calcaneal spur and plantar fasciitis are the most common causes of plantar heel pain. From clinical point of view, both these pathologies have different issues. Plantar fasciitis is most commonly caused by overuse or damage to the ligament, leading to inflammation and stiffness. Heel spurs are most commonly caused by bruising or damage to the heel bone, causing a calcium deposit to form past the edge of the bone. Usually, patients have combined intense calcification, overgrown calcaneus bone, and plantar fascia tendinopathy. However, in some cases, these disorders are isolated [16,17].

Plantar fasciitis is a common and often impairing condition that requires appropriate treatment, including conservative (lifestyle modification, stretching, orthotic devices, extracorporeal shockwave therapy), pharmacological (oral analgesia and non-steroidal anti-inflammatory drugs, steroid injections, botulinum toxin, protein-rich plasma), as well as surgical (endoscopic surgery) intervention when patients do not respond to conservative methods. While 80% of patients with heel pain have plantar fasciitis, there are many other differential diagnoses [18]. Johal and Milner [19] demonstrated a significant association between plantar fasciitis and heel spur formation. Menz et al. [20] reported that heel spurs and thickening of the plantar fascia often coexist in individuals with heel pain. It was concluded that isolated heel spurs are rare and that tenderness on heel palpation does not appear to differentiate these conditions.

It should be emphasised that the effects of the discussed physical method on the treatment of heel spurs with plantar fasciitis are much less verified. So far, there are no clear recommendations confirming the effectiveness of HILT procedures in heel spur and plantar fasciitis management. Therefore, this study aimed to evaluate the effectiveness of HILT for pain management in patients with heel spur and plantar fasciitis.

## 2. Materials and Methods

### 2.1. Design

The research project was conducted in the Clinical Research Laboratory at the Institute of Health Sciences, Opole University, Opole, Poland. The study protocol was approved by the Bioethics Committee of the Wroclaw Medical University, Poland (KB–795/2017). The study was prospectively registered in the Australian New Zealand Clinical Trial Registry platform with registration number ACTRN12618000744257 (3 May 2018). All participants gave their written informed consent to participate in the study, which was conducted following the Declaration of Helsinki and Good Clinical Practice guidelines.

### 2.2. Randomization

A research team, consisting of an internist, an orthopaedist, a radiologist, and a neurologist, qualified potential participants to participate in the research project. The procedure was conducted in outpatient service at the Institute of Health Sciences, Opole University, Opole, Poland. The assignment of participants to one of the two groups—study group or control group—was purely random, i.e., extracting the numbers from the website by a computer generator and assigning codes to individual patients, resulting in a randomised distribution of patients during the study. The person who performed the statistical analysis and the lead project manager who estimated the study outcomes received coded results and were unable to recognize the patient’s identity. They had no contact with the study participants. All measurements were performed by the same researcher (a laboratory scientist) to eliminate any bias affecting the validity of the collection of individual results. The same physical therapist also provided all treatments. The physical therapist had no contact with the eligibility team or staff analysing obtained results.

### 2.3. Participants

Patients were subject to inclusion criteria, such as diagnosed heel spur with plantar fasciitis: (1)Chronic nature of the condition in question (at least six months of symptom manifestations);(2)Persistent pain of plantar fasciitis physical examination:
a.Pain reproduced by palpating the plantar medial calcaneal tubercle at the site of the plantar fascial insertion on the heel bone,b.Pain reproduced with passive dorsiflexion of the foot and toes, andc.Windlass test—passive dorsiflexion of the first metatarsophalangeal joint—test to provoke symptoms at the plantar fascia by creating maximal stretch), positive test if pain is reproduced); and(3)A current X-ray image of the foot (heel spur).

Only adults could participate in the study. Patients with the following exclusion criteria were not enrolled in the project: diagnosed cancer, pregnancy, status post pacemaker implantation, and foreign-body implants in the area of laser radiation. Additional exclusion criteria included skin disease or history of surgery in the area of HILT application, acute infections, impaired blood coagulation, cardiac arrhythmias and conduction disorders, other foot conditions, mental disorders, sensory disorders, analgesia, and participation in supportive therapies. Other reasons for excluding an individual from participating in the study included the patient’s significantly hindered cooperation (compulsive use of drugs and psychoactive substances), taking medications with analgesic or anti-inflammatory effects, and non-experience of pain throughout the research project. Neurological and metabolic conditions were also excluding criteria.

The characteristics of the study groups in terms of age, weight, height, BMI, sex, and examined extremity was shown in Table 1. There were no statistically significant differences between the study group (*n* = 30) and the control group (*n* = 30) in terms of the listed variables. 

Following the Consolidated Standards of Reporting Trials (CONSORT) guidelines for the registered randomised clinical trials, the patient flow during the entire study period is shown in Figure 1. In both groups, all participants completed their treatment. The same was true for the assessment stage conducted one month after completion of the study. On the other hand, two group B patients were excluded at the follow-up stage after three months due to the exacerbation of pain symptoms and the need to take analgesic pharmacological agents. In contrast, all group A patients were analysed three months after the end of treatment.

### 2.4. Treatment

The patients assigned to the study group (group A) were treated with HILT using the Cosmogamma Cyborg Laser 1064 (Technomex, Gliwice, Poland), and they underwent standard physiotherapy for their condition, i.e., sonotherapy using ultrasounds (US) generated by the Intelect Advanced Combo (Chattanooga, Guildford Surrey, United Kingdom). Sonotherapy was a primary procedure, while HILT was an experimental stimulus [21,22,23].

HILT was performed using a point applicator with a 30-cm^2^, cone-shaped diffuser positioned in the calcaneal tuber region at the site of the greatest pain complaints found during the patient’s physical examination (treatments were repeated in all patients because the applicator covered the same surface). The treatment parameters were as follows: power—7 W, dose—149.9 J/cm^2^, duration—12 min, wavelength—1064 nm, duty cycle—90%, and total energy—4496 J. 

In contrast, ultrasound treatments used the following parameters: 0.8 W/cm^2^, 5 min, 1 MHz frequency, and 100% load factor for the period. A coupling substance in the form of an ultrasound gel was used for ensuring both effective conductivity of ultrasound waves and optimal contact between the transducer and the treated region. 

The patients were informed how they should prepare for the therapy, with a particular focus on how they should prepare their skin (clean and free of ointments, creams, and items that impede access to the treated region). In both cases (HILT and US), five treatments per week were performed for a period of three weeks (Monday to Friday), where treatment continuity was a prerequisite. The devices used in the present study for HILT and ultrasound therapies are presented in Figure 2.

Group B patients (control group) were treated with sham (passive) HILT therapy and active ultrasound therapy. The sonotherapy parameters were identical to those of the study group. The HILT application, on the other hand, was a sham treatment; however, all rules were consistent with the methodology typical of that treatment—technical parameters set on the device and sound signal during the treatment so that patients were not able to recognize to which group they were assigned. 

### 2.5. Measurements

Visual Analogue Scale (VAS) was used as a subjective assessment for analysing pain complaints. The scale ranged from 0 to 10, where 0 stands for “no pain” and 10 for “the greatest pain”. Each patient was asked to indicate with a slider the degree of experienced pain on the day of the study [24,25].

The Laitinen Pain Scale (LPS) was used for the subjective and point-wise analysis of the pain level according to four rates: pain intensity, pain frequency, frequency of taking analgesics, and limitation of motor activity. The patient assigns points ranging from 0–4 to each of the examined rates (0 indicates no pain-related problem, whereas 4 shows the greatest difficulty in terms of pain) [26,27].

In both groups, all measurements were taken before and after the treatment. After four and twelve weeks, follow-up measurements were used for verification of long-term effects of the therapy. Throughout the follow-up process, patients had to maintain the regimen implied by the research protocol.

### 2.6. Sample Size

The sample size of the presented study was based on group differences in primary outcomes (means and standard deviations of pain experience), which were estimated at 20 participants. A 20% loss to follow-up was allowed for in calculations. The same applies to historical information from our unit that 45% of patients offered conservative management (physical therapy agents) for heel spurs with plantar fasciitis opted for HILT within six months.

### 2.7. Statistical Analysis

The statistical analysis was performed using Statistica 13 software (TIBCO, Inc., Palo Alto, CA, USA). Arithmetic means, medians, standard deviations, quartiles, and variation range (i.e., extreme indications) were estimated to assess measurable variables. To assess qualitative variables, frequencies of their occurrence (i.e., percentages) were determined. All estimated quantitative variables were verified using the Shapiro–Wilk test to determine a type of distribution. In contrast, a comparison of qualitative variables between groups was made using the chi-square test (χ^2^). Intergroup comparisons between outcomes in samples 1, 2, 3, and 4 (M1—before treatment, M2—after treatment, M3—1 month after study completion, M4—3 months after study completion) were performed using Friedman’s Analysis of Variance and post-hoc test (Dunn’s test). The comparison of indications between the study group and control group was estimated using the Mann–Whitney U test or the *t*-test for independent samples, depending on the meeting of the conditions (normal distribution or distribution failing to meet the criteria). A significance level of α = 0.05 was used for all comparisons.

## 3. Results

The comparison of changes in pain scores obtained in four measurements (M1—before treatment, M2—after treatment, M3—1 month after study completion, M4 – 3 months after study completion) between the study group and control group by using VAS are shown in Table 2. In both groups, the mean value of the pain score changed statistically significantly (main effect: *p* < 0.05). A statistically significant decrease in the study group was observed between M1 and M2 by 3.5 pts, between M1 and M3 by 3.7 pts, and between M1 and M4 by 3.2 pts. On the other hand, the control group showed a statistically significant decrease between M1 and M2 by 3 pts, between M1 and M3 by 3.4 pts, and between M1 and M4 by 3.2 pts.

A comparison of pain scores between the study group and the control group was conducted using VAS (Figure 3). However, there was no difference in outcomes between the groups (*p* > 0.05), which indicated that the treatment was effective in both groups. Nevertheless, there was no clinical advantage of HILT over sham treatments observed. The gradual (albeit slow) recurrence of pain in long-term follow-ups—especially between 1–3 months—was also typical, demonstrating that the physical treatments did not bring any stable nor long-lasting remission. Another interesting observation is that up to one month after completing therapy, the outcomes improved to some extent (not statistically significant differences) in both groups.

The comparison of changes in pain scores obtained in four measurements between the study group and control group by using LPS are shown in Table 3. In both groups, the mean value of the pain score changed statistically significantly (*p* < 0.05). 

The comparison of pain between both groups was also conducted using LPS (Figure 4). No differences were observed between both groups at each treatment stage (*p* > 0.05). As before, there was no treatment advantage of HILT compared to the controls. Unfortunately, a slow recurrence of pain symptoms was observed in both groups in the follow-up conducted three months after completing the study. It should also be noted that there was a slight (not statistically significant) remission of pain complaints in the long-term assessment up to one month.

## 4. Discussion

In the literature (Web of Science, MEDLINE, PubMed, Physiotherapy Evidence Database), only a few publications address the discussed subject, which effectively precludes an honest and credible discussion, comparison, and potential remarks with the existing reports.

Yesil et al. [28] attempted to investigate the effectiveness of HILT and exercises in reducing pain caused by heel spurs. Forty-two individuals (in the presented study, the number of participants was originally sixty one) were enrolled in the study, and they were assigned to two comparison groups. In the first group, Yesil et al. applied HILT (age of patients—47.6 years, BMI—31.1 kg/m^2^, wavelength—1064 nm, peak power—3 kW, dose—360–1780 mJ/cm^2^, pulse duration—120–150 µs, power—10.5 W, frequency—10–40 Hz, duty cycle—0.1%, transducer diameter—0.5 cm, total energy—1281.1 J) and exercises (the duration of kinesiotherapy was approx. 25 min per day; the therapy consisted of a set of stretching, active, strengthening exercises). In the second group, quasi-HILT and exercises were applied (age of patients—43.8 years, BMI—31.3 kg/m^2^). Fifteen treatments were conducted over three weeks (the connecting element between the two projects). VAS, Roland–Morris Scale (RMS), Foot and Ankle Outcome Score (FAOS), 36-Item Short-Form Health Survey (SF-36), and a podoscope device for screening foot-pressure distribution were used for the evaluation of obtained outcomes. After 4 and 12 weeks, a control measurement was also performed (same duration of long-term follow-ups as in this project). Finally, it was noted that the measured parameters significantly improved in both groups. However, as in the presented study, no intergroup differences were shown. The report received a high methodological score of 7/10 pts on the PEDro scale.

Unfortunately, HILT did not meet our initial, promising expectations. In the presented study, no significant advantage of HILT application over standard ultrasound combined with quasi-HILT was observed. As in the study by Yesil et al. [28], the presented study included sham treatments, follow-up based on long-term results, and in-depth analyses of the pain experience. Despite some modifications in technical parameters and the use of a different device, the conclusions of this study are similar to those by Turkish researchers (in this project, sonotherapy was used as a standard treatment instead of exercises). The use of HILT is questioned for the treatment of heel spurs. 

There is also a study by Ordahan et al. [29] concerning the effectiveness of low-intensity laser therapy (LILT) compared to HILT in the treatment of only isolated plantar fasciitis; hence, the patients enrolled for the experiment do not fully reflect the study material of the presented research project (in our study were included patients with combined intense calcification, overgrown calcaneus bone, and plantar fascia tendinopathy). It should be acknowledged that the above-mentioned study scored as high as 8/10 pts on the PEDro scale. The study included seventy-five individuals who were found to have increased sensitivity to pain in the calcaneal tuber region and morning pain exacerbated with increasing load.

With the use of randomization, participants were assigned to two comparison groups. The first group (age of patients—48.73 years, BMI—31.16 kg/m^2^) was treated with LILT laser (wavelength—904 nm, peak power—240 mW, dose—8.4 J, power—0.16 W/cm^2^, frequency—5000 Hz, transducer diameter—1.5 cm^2^, total energy—680.4 J), while the second group (age of patients—48.65 years, BMI—31.22 kg/m^2^) was treated with HILT, where the therapy was divided into two stages. The first three sessions were performed during the first stage (wavelength—1064 nm, peak power—12 W, dose—6 J/cm^2^, duration—75 s, power—8 W, intermittent cycle, total energy—150 J), and the following six sessions were performed during the second stage (wavelength—1064 nm, peak power—12 W, dose—6 J/cm^2^, duration—30 s, power—6 W, intermittent cycle, total energy—120–150 J). Nine exposures were performed over a three-week period. Each physical therapy treatment was followed by stretching exercises; patients also received a silicone corrective insole. The discomfort was examined using VAS, heel tenderness index (HTI), and FAOS. Improvements in the examined rates were observed in patients in both groups; however, HILT showed significantly greater utility. 

Similar conclusions are presented in another study [21], where there was no statistically significant difference between the groups (HILT vs. LILT) according to VAS (pain in a general reduction in three weeks: 2.57 vs. 2.88). 

Therefore, Ordahan et al. [22] postulated that HILT is a more effective treatment than LILT, which further justifies using this technique in clinical practice. However, there was no reference to other acknowledged and standard practices concerning plantar fasciitis, i.e., in reference to physical rehabilitation, as in the case of study by Yesil et al. [23], or even to another physical technique, as in the case of the presented study.

### Limitations

Certainly, the weakness of this study is population size. Carrying out the study in a single, not-too-large research centre also contributed to a longer period of conducting the presented study, which would certainly not have been the case in a multi-centre project. In the future, it is important to increase the reliability based on larger population size and to have the possibility to compare outcomes to other comparison groups, i.e., other therapeutic approaches for heel spurs with plantar fasciitis, to answer the question: Which treatment will prove to be the most effective? Moreover, future research projects should not only include subjective scales, tests, and questionnaires but also objective measurement tools (surface electromyography, muscle strength assessment, isokinetic dynamometry—Biodex). Another problem concerns the establishment of uniform treatment parameters that could be verified by researchers from different centres. Nowadays, research teams use HILT’s methodology and select treatment parameters relatively freely. A uniform algorithm would be helpful.

## 5. Conclusions

The high-intensity laser therapy (HILT) used in this research project does not appear to be effective in treating pain symptoms in patients with heel spurs and plantar fasciitis compared to the conservative standard physiotherapeutic approach. Currently, HILT cannot be recommended as a helpful pain management treatment for patients with heel spurs and plantar fasciitis.

## Figures and Tables

**Figure 1 jcm-10-04891-f001:**
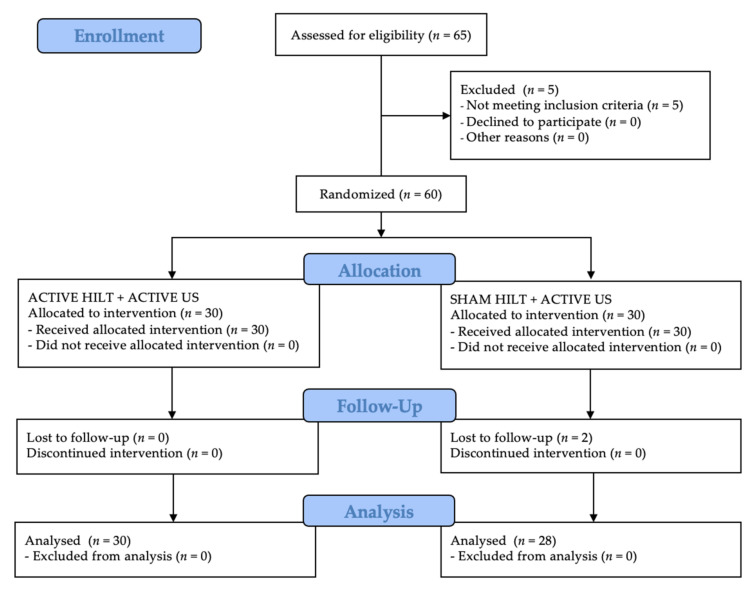
CONSORT flow chart of the study participants. Abbreviations: HILT, high-intensity laser therapy; US, ultrasound therapy.

**Figure 2 jcm-10-04891-f002:**
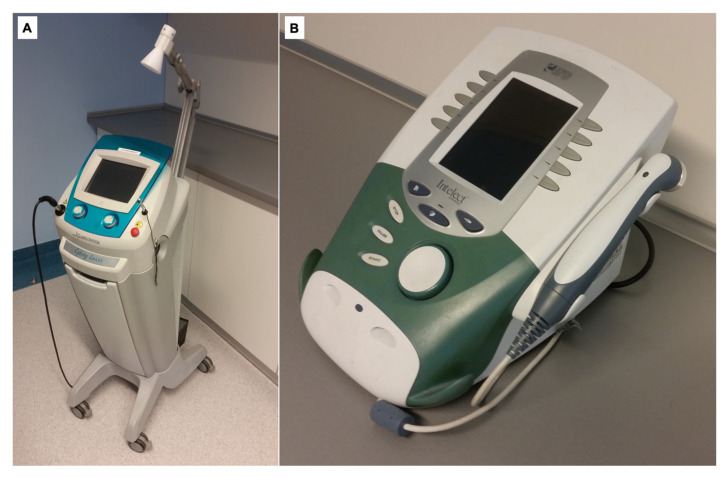
The devices for HILT (**A**) and ultrasound (**B**) treatment procedures.

**Figure 3 jcm-10-04891-f003:**
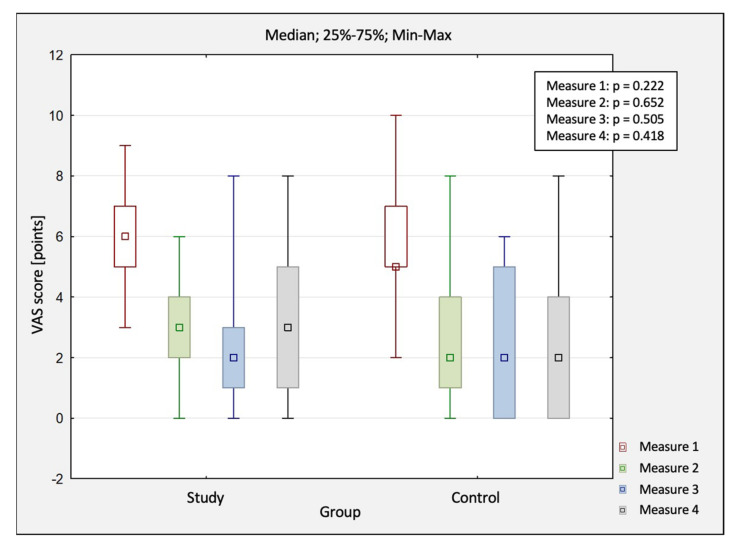
The comparison of changes in pain scores (VAS) between the study and control group. Abbreviations: M1, before treatment; M2, after treatment; M3, 1 month after study completion; M4, 3 months after study completion; VAS, Visual Analogue Scale.

**Figure 4 jcm-10-04891-f004:**
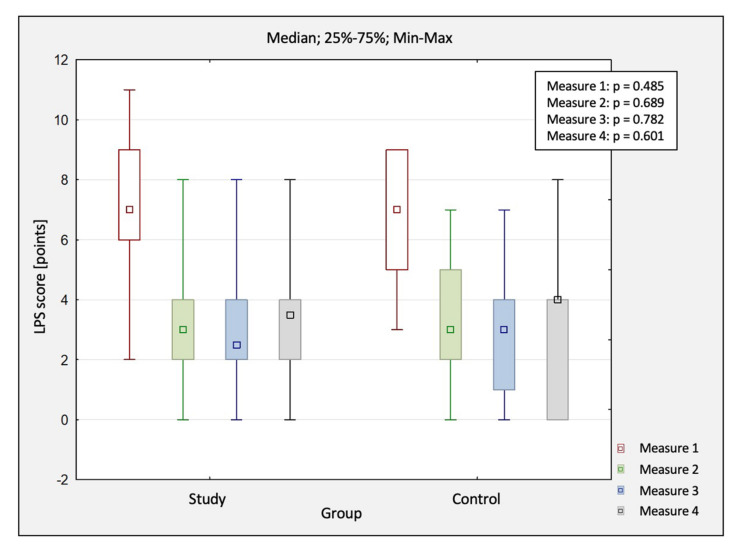
The comparison of changes in pain scores (LPS) between the study and control group. Abbreviations: M1, before treatment; LPS, Laitinen Pain Scale; M2, after treatment; M3, 1 month after study completion; M4, 3 months after study completion.

**Table 1 jcm-10-04891-t001:** Characteristics of the comparison groups.

Variable	Study Group (*n* = 30)							Control Group (*n* = 30)							*p*-Value
	$\bar{x}$	Me	Min	Max	Q1	Q3	SD	$\bar{x}$	Me	Min	Max	Q1	Q3	SD	
Age (year)	59.9	59.5	33.0	78.0	52.0	67.0	10.1	60.4	61.0	44.0	84.0	48.0	65.0	11.9	0.87 *
Weight (kg)	79.5	79.5	54.0	108	67.0	90.0	15.2	79.9	80.0	53.0	105	72.0	90.0	13.4	0.92 *
Height (cm)	171.1	170	159	187	164	178	7.7	167.8	167	152	188	160	175	10.1	0.20 *
BMI (kg/m^2^)	27.2	25.9	18.7	37.6	23.3	29.7	4.9	28.5	27.7	20.9	39.0	24.5	30.9	4.8	0.43 **
Sex	F—*n* = 19; 63.3% M—*n* = 11; 36.7%							F—*n* = 17; 56.7% M—*n* = 13; 43.3%							0.70 ***
Studied limb	L—*n* = 14; 46.7% R—*n* = 16; 53.3%							L—*n* = 16; 53.3% R—*n* = 14; 46.7%							0.68 ***

Abbreviations: *n*, number of individuals; $\bar{x}$, mean; Me, median; Min, minimum value; Max, maximum value; Q1, lower quartile; Q3, upper quartile; SD, standard deviation; F, female; M, male; L, left; R, right; BMI, body mass index. Note: * Student’s *t*-test for independent samples ** Mann–Whitney U test; *** chi-square test.

**Table 2 jcm-10-04891-t002:** The comparison of changes in pain scores (VAS) between the study and control group.

Variable	Measurement	Study Group (*n* = 30)							Control Group (*n* = 30)						
		$\bar{x}$	Me	Min	Max	Q1	Q3	SD	$\bar{x}$	Me	Min	Max	Q1	Q3	SD
VAS (pts)	M1	6.3	6.0	3.0	9.0	5.0	7.0	1.4	5.7	5.0	2.0	10.0	5.0	7.0	2.0
	M2	2.8	3.0	0.0	6.0	2.0	4.0	1.5	2.7	2.0	0.0	8.0	1.0	4.0	2.0
	M3	2.6	2.0	0.0	8.0	1.0	3.0	2.0	2.3	2.0	0.0	6.0	0.0	5.0	2.2
	M4	3.1	3.0	0.0	8.0	1.0	5.0	2.5	2.5	2.0	0.0	8.0	0.0	4.0	2.4
*p*-value *		<0.001							<0.001						
*p*-value **		M1 vs. M2: *p* < 0.001 M1 vs. M3: *p* < 0.001 M1 vs. M4: *p* < 0.001 M2 vs. M3: *p* = 1.00 M2 vs. M4: *p* = 1.00 M3 vs. M4: *p* = 1.00							M1 vs. M2: *p* < 0.001 M1 vs. M3: *p* < 0.001 M1 vs. M4: *p* < 0.001 M2 vs. M3: *p* = 1.00 M2 vs. M4: *p* = 1.00 M3 vs. M4: *p* = 1.00						

Abbreviations: *n*, number of individuals; $\bar{x}$, mean; Me, median; Min, minimum value; Max, maximum value; Q1, lower quartile; Q3, upper quartile; SD, standard deviation; M1, before treatment; M2, after treatment; M3, 1 month after study completion; M4, 3 months after study completion; VAS, Visual Analogue Scale. Note: * Friedman’s ANOVA (main effect); ** Dunn’s test (multiple comparisons).

**Table 3 jcm-10-04891-t003:** The comparison of changes in pain scores (LPS) between the study and control group.

Variable	Measurement	Study Group (*n* = 30)							Control Group (*n* = 30)						
		$\bar{x}$	Me	Min	Max	Q1	Q3	SD	$\bar{x}$	Me	Min	Max	Q1	Q3	SD
LPS (pts)	M1	7.2	7.0	2.0	11.0	6.0	9.0	2.1	6.7	7.0	3.0	9.0	5.0	9.0	2.0
	M2	3.3	3.0	0.0	8.0	2.0	4.0	1.8	3.5	3.0	0.0	7.0	2.0	5.0	2.0
	M3	3.0	2.5	0.0	8.0	2.0	4.0	2.0	2.7	3.0	0.0	7.0	1.0	4.0	2.1
	M4	3.2	3.5	0.0	8.0	2.0	4.0	2.2	2.8	4.0	0.0	8.0	0.0	4.0	2.6
*p*-value *		<0.01							<0.01						
*p*-value **		M1 vs. M2: *p* < 0.001 M1 vs. M3: *p* < 0.001 M1 vs. M4: *p* < 0.001 M2 vs. M3: *p* = 1.00 M2 vs. M4: *p* = 1.00 M3 vs. M4: *p* = 1.00							M1 vs. M2: *p* = 0.002 M1 vs. M3: *p* < 0.001 M1 vs. M4: *p* < 0.001 M2 vs. M3: *p* = 1.00 M2 vs. M4: *p* = 1.00 M3 vs. M4: *p* = 1.00						

Abbreviations: *n*, number of individuals; $\bar{x}$, mean; Me, median; Min, minimum value; Max, maximum value; Q1, lower quartile; Q3, upper quartile; SD, standard deviation; M1, before treatment; M2, after treatment; M3, 1 month after study completion; M4, 3 months after study completion; LPS, Laitinen Pain Scale. Note: * Friedman’s ANOVA (main effect); ** Dunn’s test (multiple comparisons).

## Data Availability

The data presented in this study are available on request from the corresponding author.

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
