# Peer review of "A Randomised-Controlled Clinical Study Examining the Effect of High-Intensity Laser Therapy (HILT) on the Management of Painful Calcaneal Spur with Plantar Fasciitis"

_jcm, 2021, doi:10.3390/jcm10214891_

Round 1
Reviewer 1 Report
This study investigated the effect of high-intensity laser therapy-HILT in pain management in patients with plantar fasciitis. It is a RCT in which the control group was treated with sham HILT. Both groups were treated with ultra-sound treatments. For the scientific community, the finding in the ineffectiveness of LASER therapy compared to placebo may have an interesting implication on clinical practice.
Anyway, a series of considerations raised many concerns.
- In the introduction section the authors have to describe the pathologies they have treated reporting the relevant bibliographical references.
- In methods, the authors have to better clarify the diagnostic procedures used for enrollment. About 20% of patients with heel pain are suffering from other differential diagnoses than plantar fasciitis ( for example Reiter’s syndrome, osteoarthritis, abscess in the soft tissues, entrapment of the first branch of the lateral plantar nerve and others)
- Discuss the relationship between calcaneal spurs and plantar fasciitis. (see: Cutts, S., et al. "Plantar fasciitis." The Annals of The Royal College of Surgeons of England 94.8 (2012): 539-542. And: K.S. Johal, S.A. Milner, Plantar fasciitis and the calcaneal spur: Fact or fiction?, Foot and Ankle Surgery, Volume 18, Issue 1, 2012.
- Pag 5: the authors stated that: “Each patient was asked to indicate with a slider the degree of experienced pain on the day of the study”. Six lines below the authors stated: “In both groups, all measurements were taken before and after the treatment.” This means the second evaluation assessed the pain of that moment, just after the treatment, not the pain felt in the last 24 ours. Please better define the method and review the conclusions related to the M2 results.
- Pag 5: About VAS: did the authors use a 0-10 graded VAS (normally it is graded from 0 to 100) or a NRS?
- MINOR
Pag 3, Line 82/82: “2.2. Randomization. A research team consisting of an internist, an orthopedist, a radiologist and a neurologist qualified potential participants to participate in the research project”. Please define if it was an outpatient service or other, and of which Institute.
Pag 4, Figure 1. please expand the information contained in the "allocation" boxes, defining which is the experimental group and which CG and how they have been treated.
-
Pag 4, line 12/13: “Abbreviations: M1, before treatment; M2, after treatment; M3, 1 month after study completion; M4, 129 3 months after study completion; VAS, Visual Analogue Scale.” Is this sentence referred to Figure 1?
Pag 5: “measurements”: Add the proper references of VAS and of LPS.
Page 9: “Unfortunately, HILT did not meet our expectations, and the obtained results were a huge disappointment”. Too much emphasis in this sentence! Was the expectation of the authors so high? Please rewrite this sentence
Page 9: “(in our study were included patients with combined intense calcification, overgrown calcaneus bone and plantar fascia tendinopathy)”. This is the part that must be defined in the "methods" section, combined to the diagnostic imaging modalities used.
In abstract, discussion and conclusion, the authors stated that ultrasound therapy is the “standard physiotherapeutic procedure”. Please add the proper reference.
Author Response
Dear Editors and Reviewers,
We are pleased to submit our revised paper entitled A randomised-controlled clinical study examining the effect of high-intensity laser therapy (HILT) on the management of painful calcaneal spur (plantar fasciitis) to be considered for publication in your esteemed Journal of Clinical Medicine; Section: Epidemiology & Public Health; Special Issue: Physiotherapy in Muscle Pain: Current Updates from Theory to Clinical Practice; Special Issue Editor: Prof. Dr. Tomasz Halski. We ensure that the category is an original research paper in the field of Physiotherapy, Musculoskeletal Disorders, and Pain Assessment.
We would like to sincerely thank the Editorial Board and the Reviewer for sending us such a pleasant decision stating that some Major Revisions are required and our manuscript needs to be carefully improved. The submitted manuscript is extremely important for us, so the potential opportunity for publication in your prestigious Journal of Clinical Medicine would be a great honor and motivation for our team. We would like to improve this paper as much as we can according to peer-review reports. In the revised version, we addressed all suggested corrections with a point-by-point reply. We hope that the improvements will be satisfied both for Reviewers and Editorial Board.
We very much hope that our carefully prepared, point-by-point reply in this first round of revisions, appears comprehensive and proves helpful in obtaining a positive final decision accepting our paper for publication in your prestigious Journal of Clinical Medicine.
REVIEWER #1
- This study investigated the effect of high-intensity laser therapy-HILT in pain management in patients with plantar fasciitis. It is a RCT in which the control group was treated with sham HILT. Both groups were treated with ultra-sound treatments. For the scientific community, the finding in the ineffectiveness of LASER therapy compared to placebo may have an interesting implication on clinical practice. Anyway, a series of considerations raised many concerns.
- Reply: Yes, indeed. We appreciate the promising opinion connected with our paper. We strongly agree with the referee the presented study has many limitations. However, we believe after supporting corrections and the Reviewer’s advises that our manuscript will be definitely improved. Thank you so much for all comments.
- In the introduction section the authors have to describe the pathologies they have treated reporting the relevant bibliographical references.
- Reply: We would like to apologize the reviewer. You are absolutely right. Thank you for this remark. Our previous explanation in the Introduction section was unclear. Calcaneal spur and plantar fasciitis are the most common causes of plantar heel pain. From clinical point of view these both pathologies have different issues. Plantar fasciitis is most commonly caused by overuse or damage to the ligament, leading to inflammation and stiffness. Heel spurs are most commonly caused by bruising or damage to the heel bone, causing a calcium deposit to form past the edge of the bone. Usually, patients have combined intense calcification, overgrown calcaneus bone and plantar fascia tendinopathy. We decided to examine the efficiency of HILT in the participants with both disorders. We made necessary corrections in this section and other parts of whole paper (including the title of manuscript). Now, it is fine.
- In methods, the authors have to better clarify the diagnostic procedures used for enrollment. About 20% of patients with heel pain are suffering from other differential diagnoses than plantar fasciitis ( for example Reiter’s syndrome, osteoarthritis, abscess in the soft tissues, entrapment of the first branch of the lateral plantar nerve and others)
- Reply: We added the complete and detailed diagnostic management process. Please see details in the Methods section, subheading 2.3. Participants, points 1-3.
- Discuss the relationship between calcaneal spurs and plantar fasciitis. (see: Cutts, S., et al. "Plantar fasciitis." The Annals of The Royal College of Surgeons of England 94.8 (2012): 539-542. And: K.S. Johal, S.A. Milner, Plantar fasciitis and the calcaneal spur: Fact or fiction?, Foot and Ankle Surgery, Volume 18, Issue 1, 2012.
- Reply: We discussed the relationship between calcaneal spurs and plantar fasciitis in the Introduction. Also, these suggested publications have been cited.
- Pag 5: About VAS: did the authors use a 0-10 graded VAS (normally it is graded from 0 to 100) or a NRS?
- Reply: We used the 0-10 graded VAS. The details are in the Measurements section: “Visual Analogue Scale (VAS) was used as a subjective assessment for analyzing pain com-plaints. The scale ranged from 0 to 10, where 0 stands for 'no pain' and 10 for 'the greatest pain'. Each patient was asked to indicate with a slider the degree of experienced pain on the day of the study.”
- Pag 3, Line 82/82: “2.2. Randomization. A research team consisting of an internist, an orthopedist, a radiologist and a neurologist qualified potential participants to participate in the research project”. Please define if it was an outpatient service or other, and of which Institute.
- Reply: We added the explanation to the Randomization section: “The procedure was conducted in outpatient service at the Institute of Health Sciences, Opole University, Opole, Poland.”
- Pag 4, Figure 1. please expand the information contained in the "allocation" boxes, defining which is the experimental group and which CG and how they have been treated.
- Reply: We provided additional information about interventions used in both groups in Fig. 1.
- Pag 4, line 12/13: “Abbreviations: M1, before treatment; M2, after treatment; M3, 1 month after study completion; M4, 129 3 months after study completion; VAS, Visual Analogue Scale.” Is this sentence referred to Figure 1?
- Reply: We apologies for this misunderstanding, these abbreviations and notes were removed.
- Pag 5: “measurements”: Add the proper references of VAS and of LPS.
- Reply: Done. We added necessary and relevant references:
- Scott, J.; Huskisson, E.C. Graphic Representation of Pain. Pain 1976, 2, 175–184.
- Williamson, A.; Hoggart, B. Pain: A Review of Three Commonly Used Pain Rating Scales. J. Clin. Nurs. 2005, 14, 798–804, doi:10.1111/j.1365-2702.2005.01121.x.
- Streiner, D.L.; Norman, G.R.; Cairney, J. Health Measurement Scales: A Practical Guide to Their Development and Use; 5th Edition.; Oxford University Press: Oxford, UK, 2014; ISBN 978-0-19-176545-2.
- Laitinen, J. Acupuncture and Transcutaneous Electric Stimulation in the Treatment of Chronic Sacrolumbalgia and Ischialgia. Am. J. Chin. Med. 1976, 4, 169–175, doi:10.1142/s0192415x76000214.
- Page 9: “Unfortunately, HILT did not meet our expectations, and the obtained results were a huge disappointment”. Too much emphasis in this sentence! Was the expectation of the authors so high? Please rewrite this sentence
- Reply: We rewritten this sentence in the Discussion: “Unfortunately, HILT did not meet our initial, promising expectations.”
- Page 9: “(in our study were included patients with combined intense calcification, overgrown calcaneus bone and plantar fascia tendinopathy)”. This is the part that must be defined in the "methods" section, combined to the diagnostic imaging modalities used.
- Reply: Yes, we corrected this point. Please see the Participants’ section. Thanks for the important remark.
- In abstract, discussion and conclusion, the authors stated that ultrasound therapy is the “standard physiotherapeutic procedure”. Please add the proper reference.
- Reply: Done. Please see the Treatment section
We kindly thank Reviewer 1 for her/his positive feedback and insightful comments. We would like to ensure that the entire manuscript has been carefully and thoroughly revised. All the Reviewer's comments have been considered and have contributed to improving the quality of our paper, which we would like to thank you for.
To sum up, we appreciate acceptance of our results obtained and conclusions supported by the results. Moreover, as requested, the introduction which provided a sufficient background and included all relevant references have been improved along with the research design and methods section. Also, we would like to ensure that our manuscript has been checked according to the English language and style.
Sincerely, the authors.

Reviewer 2 Report
Thank you for writing this manuscript. Even if is an interesting topic, there are certain flaws which should be corrected.
Introduction:
This subheading is weak. I would use this section to introduce what is known about plantar fasciitis (e.g., etiology, prevalence, socioeconomic costs) and this laser therapy (e.g., what is the action mechanisms, results from other research in other pathologies…).
In addition, even if is interesting the assessment of the methodological quality, outcomes analyzed, limitations and risk of bias from previous studies applied to this condition, I do not think that is adequate considering the purpose and design of this specific study. I suggest the authors to conduct a systematic review reporting properly the inclusion and exclusion criteria during the search, as well a systematic data extraction.
Finally, I think that the authors can provide a more extensive rationale for this study.
Results:
In the tables, the information is not clear at all. I do not understand the reason why the authors are reporting parametric and non-parametric descriptive data.
References:
Even if the effect of HILT was not widely assessed previously, I think the reference list supporting this manuscript is limited. I would recommend the authors (as I commented before) to expand the information about the pathology and the treatment.
Author Response
Dear Editors and Reviewers,
We are pleased to submit our revised paper entitled A randomised-controlled clinical study examining the effect of high-intensity laser therapy (HILT) on the management of painful calcaneal spur (plantar fasciitis) to be considered for publication in your esteemed Journal of Clinical Medicine; Section: Epidemiology & Public Health; Special Issue: Physiotherapy in Muscle Pain: Current Updates from Theory to Clinical Practice; Special Issue Editor: Prof. Dr. Tomasz Halski. We ensure that the category is an original research paper in the field of Physiotherapy, Musculoskeletal Disorders, and Pain Assessment.
We would like to sincerely thank the Editorial Board and the Reviewers for sending us such a pleasant decision stating that some Major Revisions are required and our manuscript needs to be carefully improved. The submitted manuscript is extremely important for us, so the potential opportunity for publication in your prestigious Journal of Clinical Medicine would be a great honor and motivation for our team. We would like to improve this paper as much as we can according to peer-review reports. In the revised version, we addressed all suggested corrections with a point-by-point reply. We hope that the improvements will be satisfied both for Reviewers and Editorial Board.
We very much hope that our carefully prepared, point-by-point reply in this first round of revisions, appears comprehensive and proves helpful in obtaining a positive final decision accepting our paper for publication in your prestigious Journal of Clinical Medicine.
REVIEWER #2
- Thank you for writing this manuscript. Even if is an interesting topic, there are certain flaws which should be corrected.
- Reply: We strongly agree with the referee the presented study has many limitations. However, we believe after supporting corrections and the Reviewer’s advises from review will be possible to improve our manuscript. Thank you so much for all comments.
- Introduction: This subheading is weak. I would use this section to introduce what is known about plantar fasciitis (e.g., etiology, prevalence, socioeconomic costs) and this laser therapy (e.g., what is the action mechanisms, results from other research in other pathologies…).
- Reply: Thank you very much for this suggestion. We significantly improved the Introduction section, also considering the recommendations of the Reviewer 1.
- Introduction: In addition, even if is interesting the assessment of the methodological quality, outcomes analyzed, limitations and risk of bias from previous studies applied to this condition, I do not think that is adequate considering the purpose and design of this specific study. I suggest the authors to conduct a systematic review reporting properly the inclusion and exclusion criteria during the search, as well a systematic data extraction.
- Reply: Please see the changes in the following section. We hope it will be acceptable for the reviewer. We agree with the Reviewers suggestion and in the nearest future (separate paper, when some more appropriate RCT papers will be published) we are going to perform high quality, according to the methodological aspects (PEDro, GRADE, Cochrane qualitative assessments), systematic review and evidence synthesis. We would like to point out that the present study was original research and we wanted to focus entirely on our findings, while this mini-review in the Introduction its’s just to strengthen the rationale for our research.
- Introduction: Finally, I think that the authors can provide a more extensive rationale for this study.
- Reply: The main rationale was that there is very limited evidence in terms to well-designed and conducted RCT’s on the effectiveness of HILT on plantar fasciitis and heel spur, which was discussed in the 6th paragraph: of the Introduction section.
- Results: In the tables, the information is not clear at all. I do not understand the reason why the authors are reporting parametric and non-parametric descriptive data.
- Reply: We wanted to present all necessary and significant data. We believe all of them would be interesting for potential readers. We kindly would like to ask the referee to accept our point of view. In our opinion, both tables presenting the results, Tab. 3 for VAS and Tab. 4 for LPS are readable and presenting most significant analyses using non-parametric tests due to the lack of a normal distribution and the low sample size, the analyses were performed: Friedman's ANOVA for main effect and Dunn's test for multiple comparisons.
- References: Even if the effect of HILT was not widely assessed previously, I think the reference list supporting this manuscript is limited. I would recommend the authors (as I commented before) to expand the information about the pathology and the treatment.
- Reply: Done, we supported the manuscript with additional and relevant references.
We kindly thank Reviewer 2 for her/his positive feedback and insightful comments. We would like to ensure that the entire manuscript has been carefully and thoroughly revised. All the Reviewer's comments have been considered and have contributed to improving the quality of our paper, which we would like to thank you for.
To sum up, we appreciate acceptation of a research design and methods section which has been adequately described. We would like to ensure that the results obtained and conclusions supported by the results have been slightly improved. However, as requested, the introduction which provided a sufficient background and included all relevant references have been carefully improved along with the research design and methods section. Also, we would like to ensure that our manuscript has been checked according to the English language and style.
Sincerely, the authors.

Round 2
Reviewer 1 Report
The corrections made to the text are satisfying
Author Response
We would like to thank Reviewer 1 for her/his detailed revisions and constructive comments which finally improved the quality of our manuscript.
With kind regards, the authors.
Reviewer 2 Report
The authors improved significantly the introduction subheading and provided a more appropriate background about laser therapy and heel pain and rationale.
As I commented before, I consider that this "mini-review" reported in Table 1 is not appropriate. Systematic reviews are required to discuss a complex and detailed methodological procedure for the data search strategy (not only the databases used), methodological quality assessment (reporting each item of the PEDro scale), study eligibility, study appraisal and data extraction (which should be systematic according with the study design of the included articles)... As these points are not reported (and should not be since this is an introduction for a clinical trial), the mini-review could be considered as biased and could be confusing for readers thinking that the mini-review was systematically conducted. My recommendation is to delete the table and keep the information from a different point of view.
Author Response
We agree with the Reviewer's point of view and as recommended, Table 1 has been definitely removed from the Introduction. The remaining tables were renumbered as Tabs 1-3, both in the headings and in the text. We also improved the last, 6th paragraph of the Introduction. We hope that our revisions will meet the Reviewer's expectations.
We would like to thank Reviewer 2 for her/his detailed revisions and constructive comments which finally improved the quality of our manuscript.
With kind regards, the authors.